



# On the calibration of FIGAERO-ToF-CIMS: importance and impact of calibrant delivery for the particle phase calibration

Arttu Ylisirniö[1], Luis Barreira[1,2], Iida Pullinen[1], Angela Buchholz[1], John Jayne[3], Jordan E. Krechmer[3],
Douglas R. Worsnop[3], Annele Virtanen[1], and Siegfried Schobesberger[1]

[1]Department of Applied Physics, University of Eastern Finland, FI-70211 Kuopio, Finland.

[2]Atmospheric Composition Research, Finnish Meteorological Institute, Helsinki, Finland.

[3]Center for Aerosol and Cloud Chemistry, Aerodyne Research, Inc., Billerica, MA, USA.

*Correspondence to*: Arttu Ylisirniö (arttu.ylisirnio@uef.fi)

## Abstract

The Filter Inlet for Gases and AEROsols (FIGAERO) coupled with a Time-of-Flight Chemical Ionization Mass Spectrometer (ToF-CIMS) enables online measurements of both gas-phase and particle phase chemical constituents of ambient aerosols. When properly calibrated, the incorporated particle filter collection and subsequent thermal desorption enable the direct measurement of volatility of said constituents. Previously published volatility calibration results however differ from each other significantly. In this study we investigate the reason for this discrepancy. We found a major source of error in the widely used syringe deposition calibration method that can lead to an overestimation of saturation vapour pressures by several orders of magnitude. We propose a new method for volatility calibration by using atomized calibration compounds that more accurately captures the evaporation of chemical constituents from ambient aerosol particles. We justify our claim with evaporation modelling and direct Scanning Electron Microscopy imaging, while also presenting possible error sources of the atomizer method. We finally present how typical calibration parameters derived with both methods impact the Volatility Basis Set (VBS) derived from measurements of Secondary Organic Aerosols (SOA).

## 1 Introduction

Organic aerosol (OA) has received substantial attention during the past decades due to its large fraction of the total atmospheric aerosol mass around the globe (Hallquist et al., 2009; Jimenez et al., 2009). The tendency of the organic matter to stay in the particles or evaporate is dictated by the volatility of the OA constituents. This information is also critical for atmospheric models for accurate treatment of secondary organic aerosols (SOA) in the these models (Sporre et al., 2020). During the past years, several techniques have been developed to measure the physicochemical properties of the OA, including volatility. A major class of these techniques relies on heating the aerosol particles followed by compositional analysis of the evaporating



molecules by mass spectrometers. Examples of these techniques are the Thermal-Desorption Chemical Ionization Mass Spectrometer (TD-CIMS, Smith et al., 2004), the Micro-Orifice Volatilization Impactor Coupled to a Chemical Ionization

Mass Spectrometer (MOVI-CIMS, Yatavelli and Thornton, 2010) and the Chemical Analysis of Aerosols Online – Proton Transfer Reaction Mass Spectrometer (CHARON – PTR-MS, Eichler et al., 2015). Another technique, which has gained popularity, is the Filter Inlet for Gases and AEROsols (FIGAERO) coupled with Time-of-Flight Chemical Ionization Mass Spectrometer (ToF-CIMS). Originally introduced by Lopez-Hilfiker et al., (2014), this technique has been employed in numerous field and laboratory studies (e.g. D'Ambro et al., 2017; Breton et al., 2018; Isaacman-Vanwertz et al., 2018; Riva

et al., 2019). The FIGAERO inlet enables semi-continuous gas-phase and particle-phase measurements of aerosol. The latter is done via filter collection followed by heating of the collected aerosol particles and simultaneous sampling of desorbing compounds, which can be identified and quantified by the ToF-CIMS. It also enables extraction of volatility information of the particle phase through the investigation of thermograms: the measured signal as a function of linearly ramped desorption temperature. In particular, the temperature of peak signal ($T_{max}$) has turned out to be a useful measure (see Sect. 2.5 for details).

When accurately calibrated, these measured $T_{max}$ values can be directly related to saturation vapor pressure values ($P_{sat}$) and used to estimate the volatility of the chemical constituents in the aerosol particles (Lopez-Hilfiker et al., 2014). However, only a considerably small amount of studies has taken advantage of this possibility and have reported the calibration procedures used for quantifying the relationship between $T_{max}$ and $P_{sat}$ (Lopez-Hilfiker et al., 2014; Stark et al., 2017; Bannan et al., 2019; Joo et al., 2019; Nah et al., 2019; Ye et al., 2019). Fig. 1 reproduces the published calibration results known to us, for a direct

comparison. It illustrates remarkable discrepancies between individual calibration results. One issue here is that $P_{sat}$ values used in the calibration fits, either literature-based or model-derived, vary significantly between studies (by up to 4 orders of magnitude for the same compound, see Table S1). These discrepancies stems in part from notorious difficulties in measuring and estimating the saturation vapor pressure of low-volatility compounds. Bannan et al., (2019) proposed a solution to this problem by using series of polyethylene glycol (PEG) compounds, which showed good agreement sof measured $P_{sat}$ values

between different experimental methods (Krieger et al., 2018). Other issues may arise from differences in the exact calibration methods. All calibration lines shown in Figure 1 have been produced by depositing known amounts of calibration compounds in solution on the FIGAERO filter using a micro syringe (later referred as syringe method). However, there is a remarkable wide variation in the level of detail at which published calibrations have been described; specifically, in terms of used solvent, solution concentrations and amount of material deposited onto the filter.


In this study we investigated the possible reasons for the large discrepancies between many reported calibration lines (Fig. 1). We repeated the calibration measurements described in Bannan et al., 2019 with PEG (4-8) compounds, and a set of carboxylic acids, and probed the effect of different solution concentrations on the calibration results. As the FIGAERO inlet itself is initially designed to study aerosol particles, we further conducted calibration experiments via atomizing the calibration

compounds (later referred as atomizer method) and found remarkable differences compared to the experiments performed via micro syringe depositions. Furthermore, as several studies performed with FIGAERO-ToF-CIMS use the syringe method also



to calibrate for the sensitivity of the instrument (Liu et al., 2016; Breton et al., 2019), we compared the two previously mentioned methods also in terms of sensitivity calibrations. We furthermore investigated potential impacts of different heating ramp rates and aerosol particle sizes to results using the atomizer method. In light of our results, with further support from evaporation modeling and direct scanning electron microscopic (SEM) measurements, we propose that atomizer method should from now on be used for calibrating the FIGAERO-ToF-CIMS volatility range.

## 2 Methods

### 2.1 FIGAERO-ToF-CIMS

The operation of the FIGAERO inlet is thoroughly explained in previous publications, with the original inlet design described in Lopez-Hilfiker et al., 2014 and a commercialized design by Aerodyne Research, Inc. described in Bannan et al., 2019. In short, the FIGAERO inlet enables measurements for both particle-phase and gas-phase constituents by the use of two separate pin holes leading into the mass spectrometer. While the gas phase is sampled through one pin hole, the other is kept closed and aerosol particles are simultaneously sampled onto a PTFE filter (Zefluor, Pall Corp. 2 μm pore size). After sufficient particle mass has been collected onto the filter, the filter is moved in front of the second pin hole and the gas phase pin hole is blocked. Chemical constituents are then evaporated from the filter into the mass analyzer by a nitrogen flow that is gradually heated, ramping at a constant rate from room temperature to 200 ˚C, as measured just above the filter. The rate of the heating ramp is adjustable, and for this study, we used heating rates of 11.4 and 6.25 K min$^{-1}$ corresponding to ramping times of 15 and 30 minutes. In this study we used the commercial version produced by Aerodyne Research Inc. for the solution concentration and heating ramp rate experiments and a custom design with small deviations from the commercial version (different nitrogen flow heating system and smaller distance between the two pinholes) for the sensitivity and particle size experiments.

The ToF-CIMS (Tofwerk AG, Aerodyne Research, Inc.) was operated with an iodide-ionization scheme (Iyer et al., 2017; Lee et al., 2014) and at a mass resolution of 4000-5000. Iodide ions were generated by passing an ultrapure nitrogen flow of 1 SLPM over a permeation tube containing methyl iodide ($CH_3I$) and through a commercial Po-210 source (Model P-2021, NRD Static Control LLC) into the Ion Molecule Reaction chamber (IMR) of the instrument. The IMR was operated at a pressure of 100 mbar which was actively controlled.

### 2.2 Sample preparation

In this study, polyethylene glycols (PEG, Polypure AS) and carboxylic acids were used as standards to test the effect of solution concentration on the results. Acetonitrile (ACN, Fisher Scientific 99.8% purity) was chosen as a solvent for stock solutions since it does not react with any of the compounds used in the study whereas for example methanol, the most commonly used solvent, was found to polymerize PEGs into higher order polymers. The used PEG standards were from PEG-4 to PEG-8. The





used organic acids were pimelic acid (Sigma Aldrich, 98% purity), azelaic acid (Sigma Aldrich, 98% purity), sebacic acid (Sigma Aldrich, 99% purity), palmitic acid (Sigma Aldrich, 99% purity), oleic acid (Sigma Aldrich, 99% purity) and stearic acid (Sigma Aldrich, 95% purity). Both stock standard solutions of individual components and mixtures of studied analytes were prepared and tested.

### 2.3 Syringe deposition method

In the syringe deposition method, a known amount of the prepared standard solutions was deposited onto the FIGAERO filter via a microliter syringe (10 μl, Hamilton Co.). To access the filter, the filter holder tray was pushed out from the body of the inlet until the filter was exposed. The amount of deposited calibration standards was calculated from the solution concentration and volume of deposited solution. After deposition, the solvent is assumed to quickly evaporate from the filter, leaving behind the less volatile calibrant analyte. An illustration of the method is shown in Fig. S1 a). Solution concentrations for the syringe deposition method were 0.1, 0.01 and 0.003 g L$^{-1}$ for the PEGs and 0.5, 0.1 and 0.01 g L$^{-1}$ for the acids. The deposited volume of standard solution was 1 μl, which provided a sufficient calibrant mass on the filter to ensure a clear signal. The mass deposited varied between 9 ng and 500 ng, depending on the used concentration. For sensitivity calibrations, PEG-7 standard solutions (0.01 g L$^{-1}$) were used. The deposited volume was 1-5 μl, which corresponded to a deposited mass of 10-50 ng.

### 2.4 Atomization method

In atomization method, PEG standards were prepared in an initial concentration of ~0.5 g L$^{-1}$ each in a mixture in acetonitrile (see Sect. 2.2). For delivering the calibrants to the filter, the solution was then atomized with a commercial atomizer (TSI Aerosol generator model 3076). Atomized particles were passed through a dilution volume and were continuously monitored with a Scanning Mobility Particle Sizer (SMPS, TSI model 3082 platform coupled with a TSI model 3775 Condensation Particle Counter, CPC). The dilution volume ensured that all solvent had completely evaporated from the particles before size measurement/classification and filter collection. We studied both polydisperse (mode diameter ~60nm) and monodisperse aerosol particles. Monodisperse particles were size selected from the polydisperse aerosol population with a Differential Mobility Analyzer (DMA). Schematics of the respective calibrant delivery setups are shown in Figure S1 panels b) and c).

Before the actual filter collection, the particles were passed through the aerosol collection port of the FIGAERO-inlet to maintain constant flow conditions in the setup while the collecting filter was in the desorption position and flushed with room air temperature nitrogen. When the particle concentration had stabilized, the filter was moved into the aerosol flow and the collection started. The amount of collected material was calculated based on particle size (determined by SMPS, assuming that all particles were spherical), CPC particle counts, collection time, and flow rate through the filter.

As the solvent used in the atomization method actively flushed the walls of the atomizer, dissolving any dissolvable material from to walls to the solution, it was essential to thoroughly clean the atomizer before measurements. The atomizer was also





periodically used with pure solvent and the output was monitored with SMPS to ensure that all measured particles consisted purely of calibration compounds.

During the atomization progress, the initial solution concentration slowly increases as part of the solvent evaporates inside the atomizer. However, this change of concentration only impacts the size distribution of the formed aerosol particles, which was
continuously monitored. In the atomizer method measurements, collected mass loading on the filter ranged from 100 to 200 ng. Typical collection times ranged from 10 s (polydisperse sample) to few minutes (monodisperse sample). For sensitivity calibrations performed with PEG-7, 100 nm particles with a similar mass loading range were used with initial atomizer solution concentration of 0.5 g L$^{-1}$. The measurement setup shown in Fig. S1 c).

**2.5 Data analysis, $T_{max}$ determination and calibration line fitting**

All ToF-CIMS data was pre-processed with Tofware (version 2.5.11 including FIGAERO plugin, Aerodyne Research, Inc.) running in the Wavemetrics Igor 7 programming environment and further postprocessed with custom MATLAB scripts (The MathWorks, Inc.).

For obtaining $T_{max}$ values from the thermograms, the data was first smoothed by fitting an asymmetrical lognormal function
across the peak of each thermogram. The assigned $T_{max}$ values corresponded to the maxima of these functions (Figure 2 panel a)).

Obtained $T_{max}$ values were fitted against natural logarithm of $P_{sat}$ literature values, which leads to a near-linear relationship

$$ln(P_{sat,lit}) = aT_{max} + b, \qquad (1)$$


where $a$ and $b$ are fitted parameters. Saturation vapor pressure values for any measured compound ($P_{sat,meas}$) can then be estimated by

$$P_{sat,meas} = exp^{a\,T_{max,meas}\,+b}, \qquad (2)$$


where $T_{max,meas}$ is the measured $T_{max}$ of the compound. In the field of organic aerosol studies, it is customary to express volatility in terms of saturation concentration ($C^*$). Saturation vapor pressures can be converted to saturation concentration following the ideal gas law:

$$C^*(\mu g\ m^{-3}) = \frac{P_{sat,meas}M_w}{R\,T}\,10^6, \qquad (3)$$



where $M_w$ is the molecular weight of the compound (in units of g mol$^{-1}$), $R$ is the universal gas constant (8.314 J mol$^{-1}$ K$^{-1}$) and $T$ is the temperature for which the original $P_{sat,lit}$ values were determined (in units of K; typically, as in our case, for 298 K).

As both $P_{sat,lit}$ and $T_{max,meas}$ can have significant uncertainties, an appropriate fitting method should be chosen that accounts for errors in both variables. In this study, we used the bivariate least squares method (York et al., 2004), which was implemented in MATLAB as shown in Pitkänen et al., (2016). When uncertainties were not available, as was the case with lines in Fig. 1, Deming regression was used. For a thorough discussion of linear fitting methods while taking into account measurement uncertainties in both variables, see Mikkonen et al., (2019).

**2.6 Evaporation model description**

In this study, we also compare experimental results with the simulation results of a model that was designed to interpret FIGAERO-ToF-CIMS observations. The model is described in detail in Schobesberger et al., 2018. It simulates the molecule-wise evaporation of aerosol particles from the FIGAERO filter in a clean nitrogen flow, using a modified form of the Hertz-Knudsen equation. Accordingly, peak-shaped thermograms arise from the linearly ramped sample heating due to the fast
increase of $P_{sat}$ (and $C^*$) with temperature (Clausius-Clapeyron relation). The model demonstrated how $T_{max}$ depends near-linearly on $\log(P_{sat})$ as the enthalpy of vaporization generally increases with decreasing $P_{sat}$, in agreement with observations (cf. Fig. 1). The model also allows for including interactions between desorbed vapours and instrument surfaces, which can lead to an increase in $T_{max}$, as well as non-ideal heating, which broadens simulated thermograms and adds tailing, hence potentially better reproducing observed thermogram shapes.

**2.7 Scanning Electron Microscope pictures**

To gain information about the difference between atomizer collection and syringe deposition, we took Scanning Electron Microscope (SEM) pictures of the FIGAERO filters with PEG deposited onto the filter with either method. The employed instrument consisted of a Sigma HD Variable Pressure Field Emission Gun – SEM (VP FEG-SEM, Carl Zeiss NTS, Cambridge, UK) with a Variable Pressure Secondary Electron (VPSE) detector using an acceleration voltage of 15 kV. The
pictures were taken in a 20 Pa nitrogen atmosphere. As SEM pictures are taken in very low pressures, we deposited only PEG-8 to the filter as it had the lowest vapor pressure of the used PEGs and was thus least likely to evaporate in the vacuum during the imaging.

For preparing the FIGAERO filters for the SEM when using the syringe method, we attached  the filters to a horizontal sample
holder using double sided carbon tape and deposited the volume in the middle of the filter in same fashion as in normal syringe method measurement, before moving the holder into the SEM vacuum chamber. For investigating deposition using the atomizer method, we collected 300 nm monodisperse particles into the filter for 20 minutes after which the filter was attached to an identical sample holder and moved into the SEM within 15 minutes after the collection.



# 3 Results and discussions

## 195    3.1 Solution concentration effect

We examined a range of solution concentrations for the syringe deposition method, with both PEGs and carboxylic acids. PEGs were measured as individual solutions and as a mixture. Carboxylic acids were measured as a mixture. With PEGs, we did not observe significant difference in $T_{max}$ values between mixture and individual solutions. Fig. 3 shows a shift of measured $T_{max}$ to higher temperatures with increasing solution concentration, both for PEG compounds (Fig. 3a) and for
carboxylic acids (Fig. 3b). The shown $T_{max}$ values are averages of three repetitions. Exact values with standard deviations are shown in Table S2 and Table S3. Fig. 3a also includes reported $T_{max}$ values from Bannan et al., (2019) as a reference, as they used PEG compounds with solution concentrations of ~2 g $L^{-1}$. Both panels also show the $T_{max}$ values we measured with the atomizer method, which yield the lowest $T_{max}$ values with both sets of compounds.


The results shown in Figure 3 clearly show a dependence of measured $T_{max}$ value on solution concentration deposited by syringe, with higher concentrations leading to higher $T_{max}$, whereas lowest $T_{max}$ values are measured when using the atomizer. Even though $T_{max}$ values from the lowest solution concentration of 0.003 g $L^{-1}$ in PEG measurements approach the atomizer results, there is still a difference of ~15 ˚C between the results. This difference would manifest in 1-2 orders of magnitude in
difference in estimated saturation pressure. Note that PEG-4 was not visible in the mass spectrometer data for the atomizer and the lowest solution concentration measurements. We suspect that its evaporation from the filter and from the particles is so rapid in these cases, that it has already evaporated before the start of the measurements, or, in other words, that its hypothetical $T_{max}$ lies below or too close to room temperature. This is in line with the relatively high vapor pressure of PEG-4. Its $\log_{10}(C^*)$ value of 3.12 groups it into the class of Intermediate Volatile Organic Compounds (IVOC), as described by
(Donahue et al., 2012), which have been shown to readily evaporate from particles (Li et al., 2019; Yli-Juuti et al., 2017).

We were largely able to reproduce our measurement results using the evaporation model to simulate the evaporation of mixed PEG 4-8 particles (for simplicity assuming equal mole fractions for all PEG). Figure 4 shows decent agreement between measured and modelled $T_{max}$ values for the atomizer method, considering that the model was run with practically no free
parameters in this case. With increasing initial size of the modelled evaporating particle, the modelled $T_{max}$ shift to higher values, due to the decreasing surface-to-volume ratio. By simply adjusting that size ($D_p$), to 1.3 μm and 11 μm diameter particles, respectively, the model indeed reproduced remarkably well the $T_{max}$ values obtained with the syringe method for 0.01 g $L^{-1}$ and 0.1 g $L^{-1}$.



### 3.2. Scanning Electron Microscope pictures

Figure 5 shows SEM pictures of a FIGAERO filter, with 10 µl of pure ACN (blank test, panel a)) or 3 µl of PEG-8 in ACN solution with a concentration of 0.01 g L$^{-1}$ (panel b)) deposited on the filter. After evaporation of the solvent, the PEG-8 forms a clear ring in the position where the droplet was deposited whereas pure ACN leaves no visible mark on the filter. Figure 5 panel c) shows a magnification of the filter at the edge of the "PEG-8 ring" shown in panel b), showing how the PEG forms a layer on top and possibly also inside the filter.


Figure 6 shows magnified SEM pictures of a FIGAERO filter (note different scale compared to Fig. 5). Panel a) shows a clean FIGAERO filter without collected particles, and panel b) shows a filter with collected 300 nm particles.

Figure 5 and Figure 6 clearly demonstrate how differently the calibration material deposits onto the FIGAERO filter, depending
on which method is used. We hypothesize that a vast difference in the surface-to-volume ratio of the deposited material, as implied by the SEM pictures, is particularly crucial in explaining the differing $T_{max}$ results. We expect the molecular desorption rate of the deposit in clean nitrogen to be proportional to its total exposed surface area (Hertz-Knudsen equation; Cappa et al., 2007; Schobesberger et al., 2018). The deposit's total volume, however, is proportional to the deposited amount, i.e. broadly the same in these experiments irrespective of deposition method. Indeed, it was by building on these assumptions that the
evaporation model succeeded in reproducing the observations in Fig. 4. With the much smaller surface area of the syringe deposited material, it requires more time to evaporate all the PEG-8 than from the equivalent amount of deposited aerosol particles. This time delay directly translates to a shift to higher observed $T_{max}$ values.

### 3.3. Particle size and heating ramp rate effect in atomizer method

As $T_{max}$ values have been reported to vary in aerosol measurements (Huang et al., 2018; Schobesberger et al., 2018), we investigated how different particle sizes and FIGAERO heating ramp rates influence the measured $T_{max}$ values with the atomizer method.

We performed measurements with PEG mixture aerosol particles with monodisperse mobility sizes of 80 nm and 300 nm and
mass loadings in between 150-170 ng, while using a ramping time of 15 min (Figure 7 panel a). Note that $T_{max}$ results differ from results shown in Sect. 3.1 due to different FIGAERO inlet used here. Monodisperse particles showed a consistent difference in measured $T_{max}$ of ~7 °C between 80 nm and 300 nm particles for all PEGs, which translates to roughly half an order of magnitude difference when used to calibrate the $C^*$ space. Our evaporation model confirms this difference for all PEGs except PEG-8. The difference between different particle sizes can be explained with different surface-to-volume ratios
as was discussed in the previous section.



We observed a difference of 3-5 ˚C in measured $T_{max}$ values between heating ramp times of 15 min (11.4 K min$^{-1}$) and 30 min (6.25 K min$^{-1}$) (Figure 7 panel b). The evaporation model yields the same difference between the two heating times, even though actual $T_{max}$ values are slightly overestimated. The difference in observed $T_{max}$ values between different ramping rates

is expected. With a slower linear ramping rate, for example, more time will have passed at any momentary desorption temperature, allowing a larger fraction of molecules to have already evaporated. Consequently, the supply of molecules will become exhausted at a lower desorption temperature, which causes the peak that defines $T_{max}$.

### 3.4. Sensitivity calibration comparison

The syringe deposition method has often been used to calibrate the sensitivity of the FIGAERO particle phase measurements,

i.e. to correlate the number of measured ions to the collected material on the filter (Lopez-Hilfiker et al., 2014; Liu et al., 2016). These measurements are typically done in a similar way as described in Sect 2.2 but varying the amount of deposited calibrant by varying the amount of deposited solution. The signal of the calibration compound is then integrated over the full heating period and contrasted against the deposited mass after which a linear fit yields the instrument's sensitivity.

In Figure 8 we compare the sensitivity calibration for PEG-7, done with the syringe deposition method (blue), to equivalent measurements, done with atomized monodisperse particles instead (green). The results of the two methods are in excellent agreement, which confirms the feasibility of the atomizer method also in sensitivity calibrations. However, when using the atomizer method in sensitivity calibrations, additional precautions should be taken to ensure that all assumptions made in the mass loading calculations are valid. For example, possible particle agglomeration must be considered when atomizing high

particle number concentrations, in particular when using compounds that form solid particles at room temperature and at RH prior to FIGAERO sampling, such as ammonium sulphate or citric acid. As agglomerated solid particles are generally not spherical, as is often assumed for mass loading calculations, calculated particle mass loading on the filter can be overestimated. We therefore recommend using the syringe deposition method for sensitivity calibrations, also because the amount of required instrumentation and associated errors are much smaller.


### 3.5 $P_{sat}$ of higher order PEGs

As $T_{max}$ values of the used PEGs only reach up to ~80 ˚C, but $T_{max}$ values of ambient aerosols are reported as high as 160 ˚C

(Huang et al., 2019), it would be beneficial to extend the calibration range to higher $T_{max}$ values for more accurate calibrations. PEGs are commercially available in polymer lengths of more than 30 chains, but unfortunately available saturation pressure data only extends up to PEG-8 (Krieger et al., 2018). However, as PEGs are straight chain polymers, it could be assumed that



log($P_{sat}$) of higher order PEGs increase in a linear fashion, which is also suggested in Krieger et al., 2018. We include results of $T_{max}$ measurements using higher-order PEGs (up to PEG-16) and two tentative analysis approaches via estimating the $P_{sat}$

value of those compounds in the SI Sect. S4.

### 3.6 Impact of using different calibration methods

Figure 9 panels a) and b) show Volatility Basis Set (VBS) distributions constructed from FIGAERO desorption measurements of SOA. formed from photo-oxidation of α-pinene in a flow tube experiment. We used $T_{max}$ to $P_{sat}$ calibration coefficients acquired either via the atomizer method or via the syringe method, in the latter case with a solution concentration of 0.01 g L⁻

¹ standard solution depositions. A more detailed description of the SOA production is shown in Ylisirniö et al., 2020. Note that the heating ramp rates in these calibrations were done with faster heating ramp rate than in the SOA measurements, introducing an overall small systematic error (<1 order of magnitude in $C^*$ space). However, the presented differences are unaffected. The used calibration curves are shown in panel c) and panel d) reproduces the calibration lines of Fig. 9c, but in terms of $C^*$ for a compound with molecular weight of 200 g mol⁻¹.


Results clearly demonstrate the effect of using the syringe deposition method versus the atomization method. When using the calibration coefficients from the atomizer method, there is a shift towards lower volatilities: the amount of Low Volatile Organic Compounds (LVOC) and Extremely Low Organic Compounds (ELVOC) is increased, while Semi Volatile Organic Compounds (SVOC) compounds mostly disappear. The magnitude of this shift is presented more directly, albeit

approximately, in Fig. 9d. The difference in $C^*$ between the two calibration methods is ~1 order of magnitude at 50 ˚C, increasing to ~2.5 orders of magnitude at 100 ˚C.

The difference is strong enough to have the potential to change the aerosol growth dynamics in global climate models employing VBS distributions and could thus impact the estimation of Cloud Condensation Nuclei (CCN) numbers which in

turn leads to an underestimation of the reflected solar radiation from clouds (Sporre et al., 2020).

### 4 Summary and Conclusions

In this study we introduced an improved method for FIGAERO-CIMS volatility calibration from peak thermogram value $T_{max}$ to saturation pressure $P_{sat}$, by atomizing the used calibration compounds and compared the results to the thus far more often

used syringe deposition method. With the syringe deposition method, we found a clear effect of solution concentration on measured $T_{max}$ values (e.g., Fig. 3). This effect can lead to severe overestimation of saturation vapor pressure values when derived from measured $T_{max}$. For investigating those differences in calibration results, we also employed evaporation modelling and took direct Scanning Electron Microscope pictures of calibration compounds deposited onto the FIGAERO filter. Both



the modelling and SEM images shows that the structure and the volume of the deposited unit controls the evaporation. Syringe
deposited calibration compounds form patches of material when the solvent evaporates, whereas collected aerosol particles
stay as separate particles on the filter. The atomized particles have much higher surface-to-volume ratio compared to the
syringe deposited patches and so a similar total amount of deposit will evaporate more quickly from the filter. As the FIGAERO
inlet is designed to measure ambient aerosol particles, it stands to reason that using atomizer method will yield more
appropriate calibration results than the syringe deposition method.


In Fig. S2 we show again previously reported calibration lines shown in Fig. 1, now updated with calibration lines acquired in
this study with both the atomizer (solid green line) and the syringe method (conc. 0.1 g L$^{-1}$, solid blue line). The area between
the two solid lines encompasses almost all other reported calibration lines. It should be noted that even though the calibration
lines extend all the way to 200 ˚C, $T_{max}$ values used for the fitting are below 120 ˚C in almost all the studies.


To explore possible uncertainties in the atomizer method due to sensitivities to experimental settings, we also investigated the
effect of particle size and heating ramp rate to the measured $T_{max}$ values. We found overall differences of ~7 K between 80 nm
and 300 nm PEG particles and of ~3 K between 15 min and 30 min ramp rates. These differences translate to roughly half
order of magnitude change in saturation concentration ($C^*$) space. As the used particle size has a moderate impact on the
measured $T_{max}$ values, it is advisable to use polydisperse aerosol for calibration with particle size distributions close to the
actual aerosol size distribution that is being measured.

We also tested how the atomizer method performs against the syringe deposition method in sensitivity calibrations with using
PEG-7 as calibrant compound. The two methods produced practically identical sensitivity calibration curves when using liquid
aerosol particles. However, possible measurement errors and infrastructure requirements for the atomizer method may make
the syringe deposition method more feasible for sensitivity calibrations.

We finally compared how the use of calibration curves from the two methods impact the VBS distribution derived from SOA
formed from photo-oxidation of α-pinene. We found that using calibration parameters from the atomizer method shifted the
VBS distribution ~1-3 orders of magnitude compared to the VBS distribution derived with the syringe deposition method,
especially increasing the amount of LVOC and ELVOC compounds. This shift is strong enough to affect our understanding
and modelling results of SOA formation and dynamics and ultimately, how these processes are treated in global climate
models, potentially affecting calculated CCN values.

An essential aspect of calibrating the $T_{max}$-$P_{sat}$ relationship for FIGAERO is the use of reference $P_{sat}$ values for the calibration
compounds. As we pointed out in the introduction, the $P_{sat}$ values found in the literature for typical organic compounds have
high variations depending on the literature source (see Table S1). Therefore we strongly recommend that FIGAERO-CIMS



$T_{max}$ to $P_{sat}$ calibrations should be performed using atomized PEGs, with literature $P_{sat}$ values currently being reported in Krieger et al., 2018. We note that these $P_{sat}$ values have not been verified by other studies and are subject to corrections, but

want to point out that harmonizing further FIGAERO calibrations by using PEGs would make future FIGAERO measurements more comparable to each other. For example, volatility datasets derived from FIGAERO measurements using an atomized PEG-based calibration could be corrected with minimum efforts if more accurate $P_{sat}$ values for PEG became available, or if the available set of $P_{sat}$ values was extended to higher order PEGs.


*Data availability.* The data shown in the paper is available on request from corresponding author.

*Author contributions.* AY and SS led the paper writing, AY, LB and IP made the measurements. All co-authors participated in the interpretation of the results and paper editing.


*Competing interests.* Jordan Krechmer, John Jayne, and Douglas Worsnop work for Aerodyne Research, Inc., which commercialized the FIGAERO inlet.

*Acknowledgements.* We thank Jari Leskinen from SIB Labs facilities for assistance in SEM imaging.


*Financial support.* This research was supported by the Academy of Finland (272041, 310682, 299544) and the University of Eastern Finland Doctoral Program in Environmental Physics, Health and Biology.

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





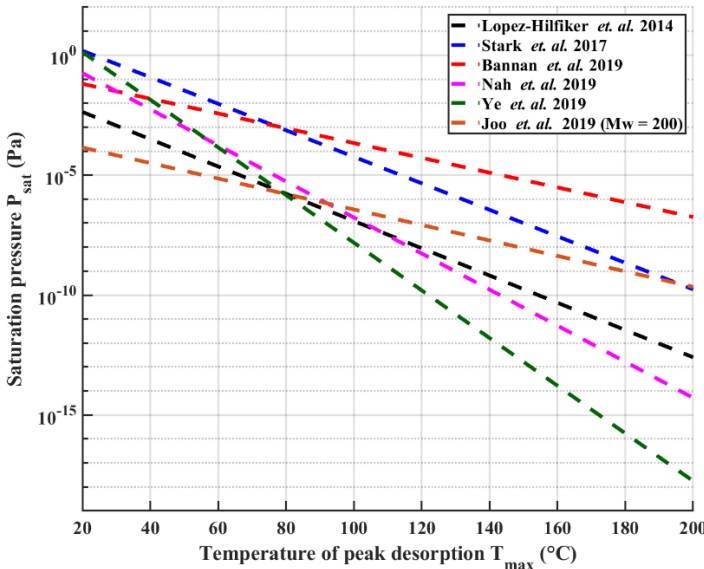


**Figure 1. Previously reported calibration measurements with temperature at peak desorption ($T_{max}$) plotted against saturation pressure $P_{sat}$. The Joo et. al., (2019) line has been converted from saturation concentration values to saturation pressure assuming a molecular mass of 200. All lines except Joo et. al., (2019) are also refitted from literature data using the fitting routine described in**
**Sect. 2.5. It is notable that in most cases the data points used for the fitting do not reach $T_{max}$ values higher than 120 °C, which is likely partially responsible for the further divergence of results when extrapolating to higher $T_{max}$.**



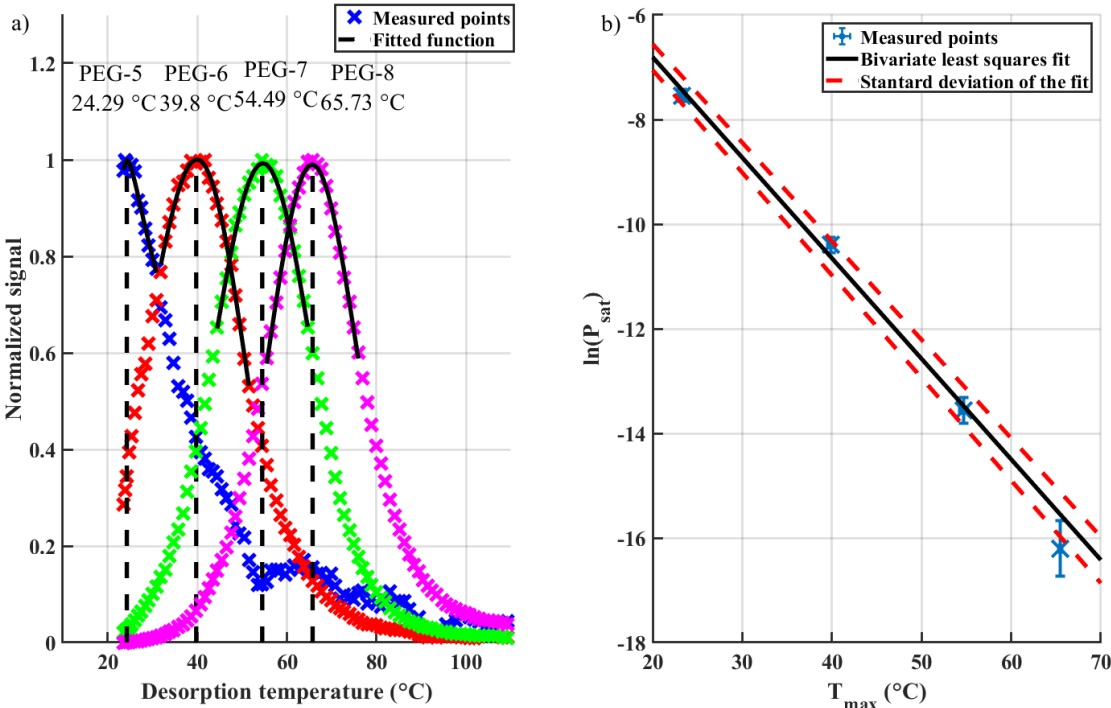

**Figure 2. Panel a) example for acquiring $T_{max}$ values from thermograms with fitted asymmetric lognormal function. Panel b) Fitting a line to the natural logarithm of literature-based saturation vapor pressures ($P_{sat}$, in units of Pa) as a function of corresponding FIGAERO-derived $T_{max}$ values, while taking the uncertainties into account. With $P_{sat}$, uncertainties are taken from the literature and with $T_{max}$, the uncertainties are defined as the standard deviation of three measurements. Fitting parameters of the line fit were $a = -0.1923 \pm 0.0039$ and $b = -2.9589 \pm 0.17$.**


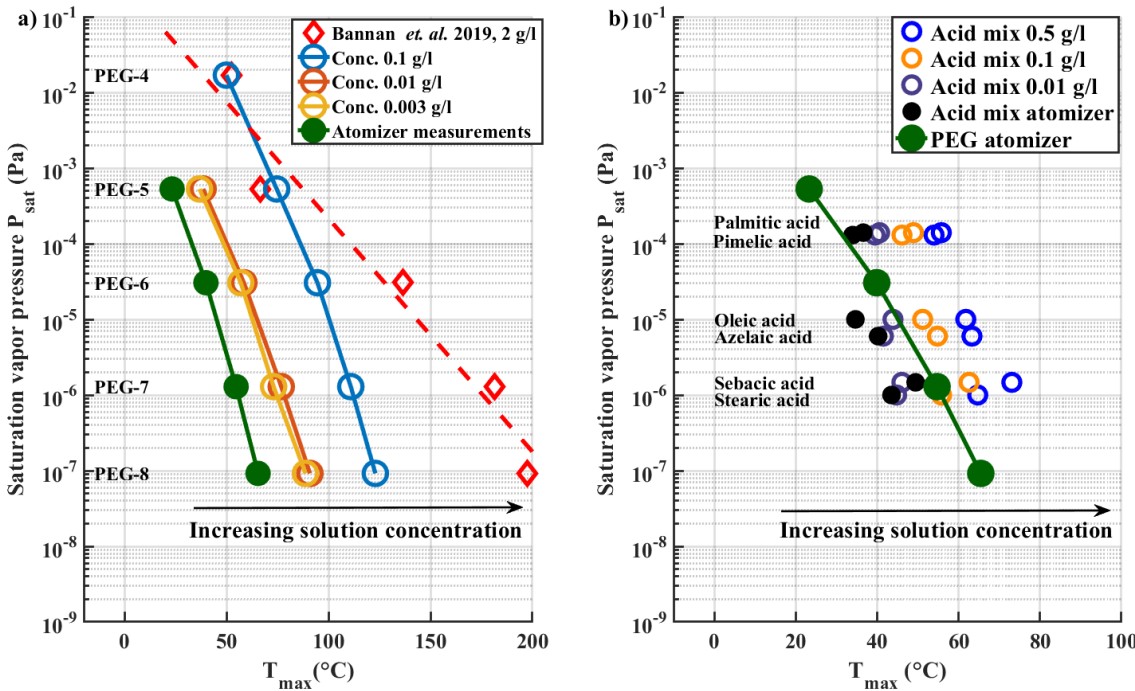


**Figure 3. Solution concentration effect. Literature-based $P_{sat}$ are plotted vs. measured $T_{max}$ for a) PEG compounds and b) carboxylic acids with a logarithmic y-axis. Black arrows in both panels indicate the direction of shift in $T_{max}$ values as the solution concentration increases. PEG atomizer results are also included in panel b) for reference.**





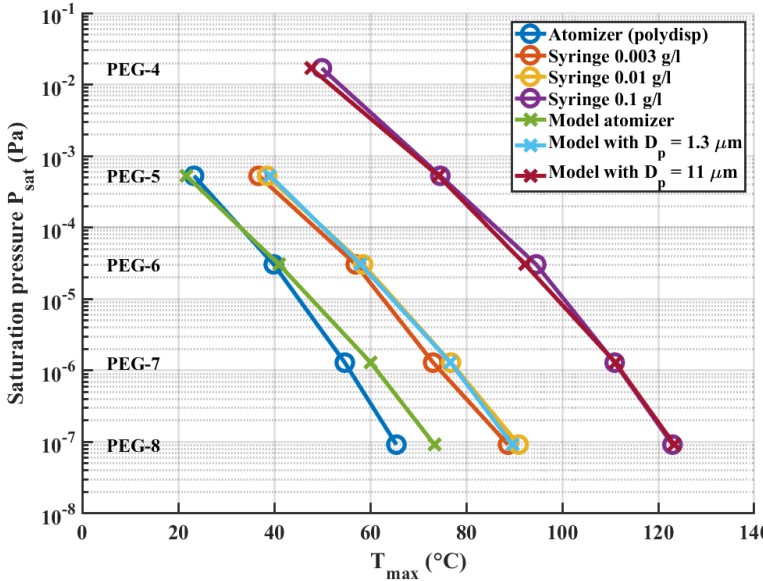


**Figure 4. Comparison of the solution concentration effect on PEG results with model results.**





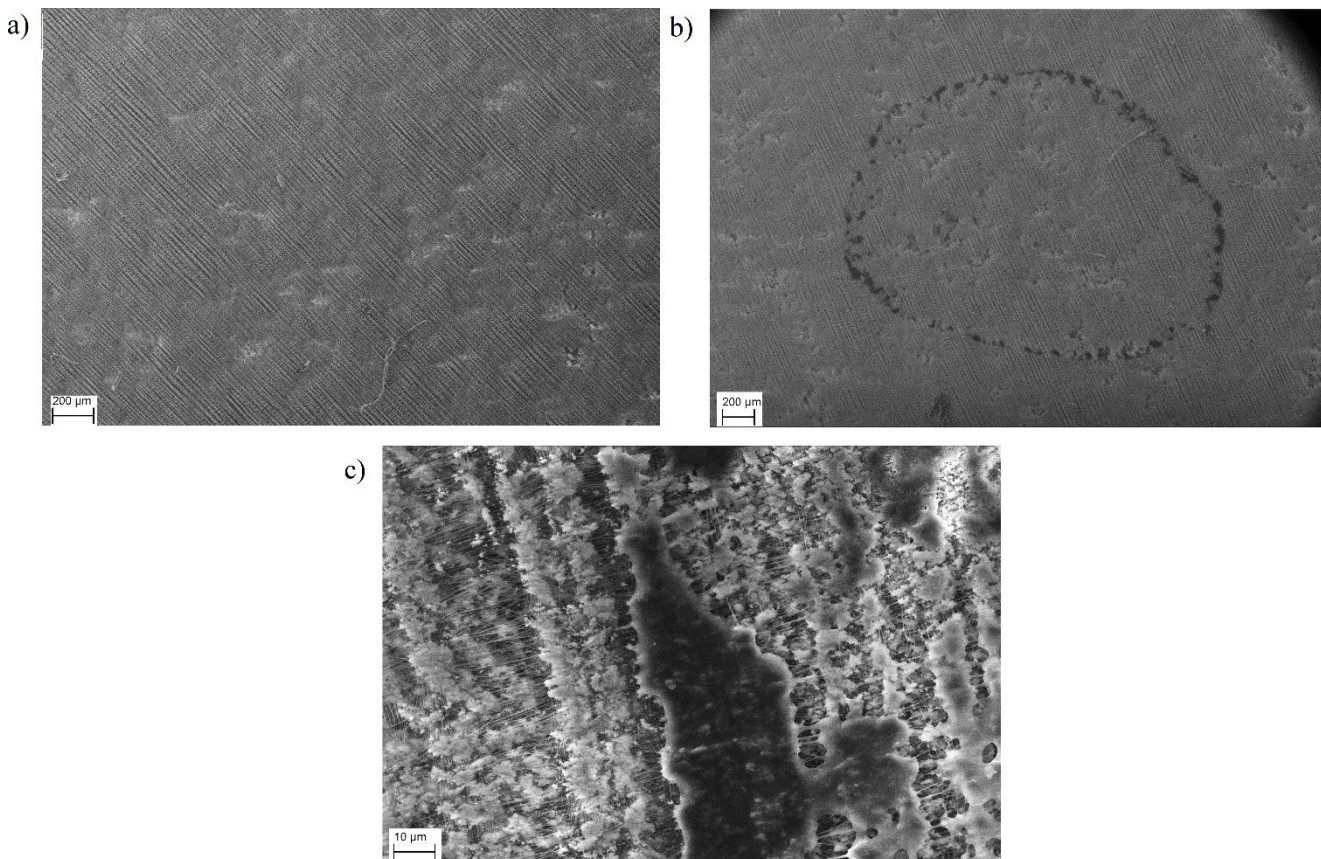

**Figure 5. SEM pictures. Panel a) 10 µl of pure ACN deposited on the filter. Panel b) 3 µl of PEG-8 with concentration of 0.01 g L$^{-1}$**
**in ACN deposited on the filter. Panel c) magnification of the edge of the PEG-8 "ring".**





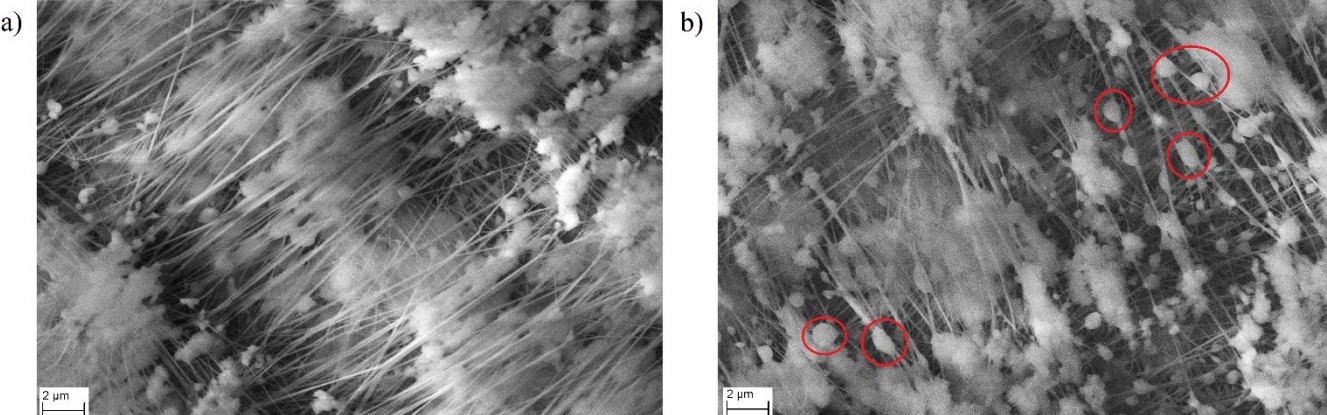

**Figure 6. SEM pictures. Panel a) magnification of a clean FIGAERO filter with no collected particles. Panel b) a magnification of a FIGAERO filter with collected 300 nm sized PEG-8 particles. Red circles in panel b) emphasize selected spots where liquid PEG-8 particles are deposited.**



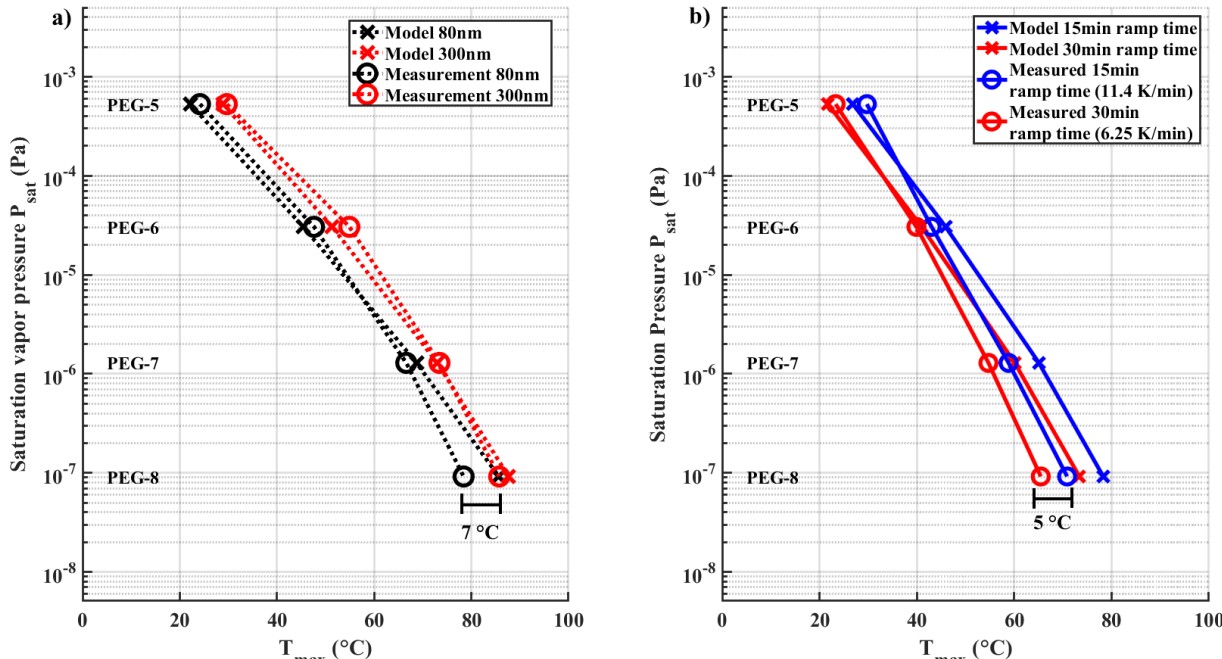

**Figure 7. a)** $T_{max}$ **values measured for 80 nm and 300 nm particles with ramp time of 15 min. The difference in** $T_{max}$ **between the two particles sizes is ~7 ˚C. Panel b) shows measured** $T_{max}$ **values of 15 min and 30 min ramping times using polydisperse aerosol particles. The difference between the two heating rates is ~ 5 ˚C.**



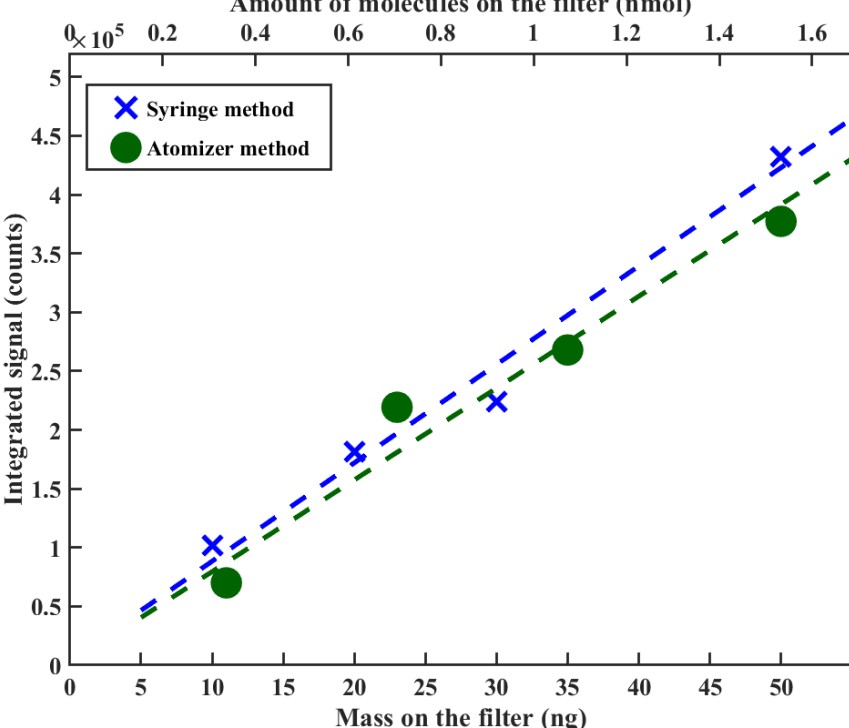

**Figure 8. Sensitivity comparison between different calibration methods using PEG-7. The bottom x-axis shows the deposited mass on the filter; the top x-axis shows the same amount in nano moles.**




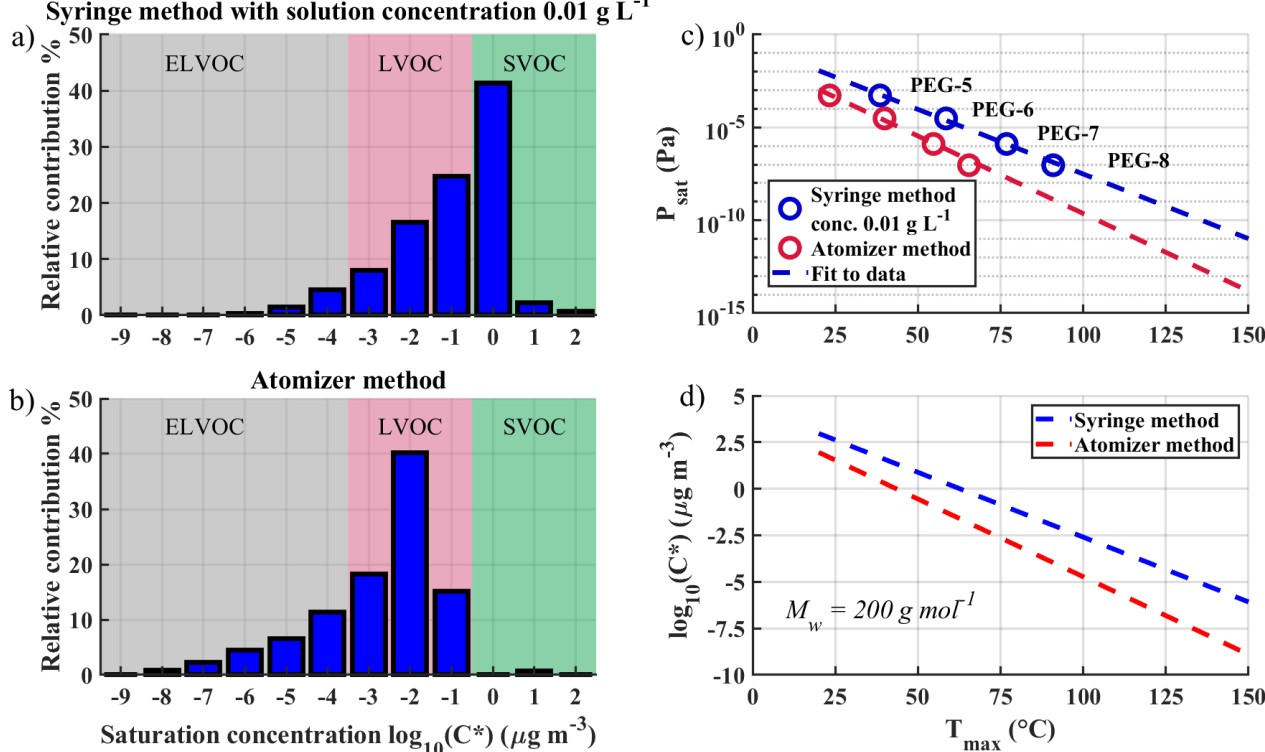

**Figure 9. Comparison of volatility basis sets (VBS) derived for the same SOA but using different calibration methods. Panel a) shows VBS determined with deposition method and panel b) shows VBS determined with atomizer method using the same data set. The respectively used calibration lines are shown in panel c). Panel d) shows how different calibration lines would impact the $\log_{10}(C^*)$ value of a compound with $M_w$ of 200 g mol$^{-1}$ with different $T_{max}$ values.**