# Peer review of "On the calibration of FIGAERO-ToF-CIMS: importance and impact of calibrant delivery for the particle phase calibration"

_Atmospheric Measurement Techniques, 2020_

## Referee Comment (RC1) · Anonymous Referee #1 · 21 Aug 2020

The authors present a comparison of volatility calibration methods for the Filter Inlet for Gases and Aerosols (FIGAERO), which is usually used to measure the particle phase composition, but can also reveal information on the volatility of the measured compounds by analyzing the maximum desorption temperature. The authors demonstrate that the widely used syringe deposition method for volatility calibration suffers from systematic errors. The paper discusses an important aspect for the small (but growing) community using FIGAERO-CIMS and is hence within the scope of AMT. It is well written and scientifically sound. I only have some remarks here and there, which should be addressed prior to publication:

**More general:**

1) Dependence on the solution concentration. The authors demonstrate that by adjusting the diameter in the evaporation model, they can reproduce the results for different solution concentrations. In order to have this in line with the subsequent reasoning about the SEM image, they should show different SEM images with different solution concentrations showing that indeed smaller structures are deposited on the filter in the case of lower solution concentrations. If the diameter really controls this behavior, also a calibration with the atomizer and particles around 1  $\mu$ m could support this.

2) The authors base their reasoning mainly on the SEM images of the FIGAERO filter for the atomization method and the syringe method. However, they only show SEM pictures for one substance. While I don't doubt their conclusion, other SEM images should be added. The authors make the statement that the vacuum in the SEM could evaporate all the other substances than PEG-8, but how can they than conclude that it is not the SEM sample preparation (i.e. bringing the filter into a vacuum), which causes the structures observed on the filter?

3) Impact of using different calibration methods. When showing the different VBS systems, I would like to see also a comparison to a VBS derived using a group contribution method or a fit to it (as e.g. in Stolzenburg et al., 2018 or Mohr et al., 2019). This would indicate which calibration method is more in line with this widely used approach, which does not rely on a direct volatility measurement.

**Minor:**

1) P.1, I.31-32: I am missing a short discussion on other volatility measurement techniques, e.g. VTDMA setups. Please add this here.

2) P.2, I.39: I am missing some laboratory studies from the CLOUD team published recently, e.g. Wang et al. (2020, Env. Sci. Techn. & Nature), Stolzenburg et al. (2018, PNAS). Also missing is Mohr et al. (2019, Nat. Commun.). In all these studies the

FIGAERO-CIMS was deployed quite successfully and they could be mentioned here for completeness.

3) P.2, I.49: Also Wang et al. (2020, Env. Sci. Techn.).

4) P.5, I.144: Did you constrain the width of the lognorm fit for the desorption? This could be necessary especially for unknown compounds, which might have isomers or fragments on the same mass yielding a bimodal structure.

5) P.6, I.185-187: If the inlet is initially at a different temperature, the supply of a constant heat rate will yield a different thermogram, as it takes longer to achieve the corresponding temperatures allowing more time for evaporation. Is this considered in the model? And how can we use calibrations performed at one temperature in comparison to measurements at different temperatures? Could the model resolve this?

6) P.7, I.202: Repeat the atomizer solution concentration to put it into the context with the syringe concentrations. Also mention here the mode diameter of the particles used for calibration or even calculate the deposited mass for this type of calibration compared to the syringe method. This would put the two methods into comparison here.

7) P.8, I.231: Instead of mentioning the different scale, I would like to see a fourth panel in Fig 4 showing the filter in the same scale as in Fig. 5c! This would help to directly compare the different structures deposed on the filter.

8) P.8, I.240: Also the larger diameters needed to explain the syringe calibration with model point into that direction. This is an important supportive argument and should be mentioned here.

9) P.8, I.251: Any hints why the different inlet behaves that way?

10) P.8, I.254: Why does it fail for PEG-8? Please elaborate on that.

11) P.10, I.295: Move "A more detailed description of the SOA production is shown in Ylisirniö et al., 2020." in front of the preceding sentence.

СЗ

12) P.10, I.299: 200 g mol-1 seems quite low for alpha-pinene HOMs, e.g. Tröstl et al. used 300 amu as mean mass.

13) P.11, I.355: Seems logical, but extremely difficult to realize in the lab. What would be the best alternative?

---

## Referee Comment (RC2) · Anonymous Referee #2 · 24 Aug 2020

General:

The authors investigated the effect of different calibration procedures for deriving vapor pressures of oxygenated organic volatile compounds by FIGAERO-CIMS measurement. They find that the structure of the dried deposited calibrants together with the heating rate affect the vapor pressure by up to a few orders of magnitude. FIGAERO-CIMS developed recently to a method with important impact on understanding the formation of secondary organic aerosols (SOA). On one hand FIGAERO-CIMS offer great opportunities for direct measurements of SOA composition and partitioning of oxygenated organic volatile compounds. On the other hand calibration and use of the

[Figure]

FIGAERO are not trivial, especially as the target compounds have low volatilities and filter sampling with thermo-desorption is involved. Despite efforts by the FIGAERO community there remain discrepancies and open questions, which are important for a larger community as FIGAERO CIMS are one of current state of the art data providers for understanding SOA and related topics. From this point of view the paper is timely and helpful. It is interesting and overall well written and clear. It addresses important issues and will add to reliability of FIGAERO-CIMS use. The paper should be published in AMT, after the authors considered one major and a few minor points.

Major comment:

The manuscript deals with calibration issues, which means it deals with quantitative issues, and it compares to results from other references. Although the authors made aware of suited fitting procedures for data with errors in y and x (line 168f) there is no detailed error analysis nor are error bars shown in the Figures 1, 4, 7, 8, S2, S3. The only errors given are the statistical errors from averaging single measurements. I find that strange for a paper that deals with quantitative analysis and urgently suggest to add more detailed error analysis' and -discussion.

Minor comments:

line 44: References should be given already here, in addition to the link to section2.5.

line 82: Are these heating rates really so accurate (2 decimal digits)? The experiments do not lead to the same maximum temperature, does that effect the integrals under the thermograms?

line 97: From PEG-4 to PEG-8. In integer steps? Please, specify precisely which PEG's you used.

line 115-120: Did you use a neutralizer ? If yes, which? How did you handle multi-charged particles in the selection by DMA. This needs to be explained, and potential errors need to be estimated and discussed.

Figure 6: I don't see a real difference between with and w/o particles. What made you think that the marked white blobs are particles? The marking with the red circles is too suggestive. I propose to leave them out. In this context: In the ambient with aqueous particles and RH, wouldn't the particles flow together and merge anyhow? I would still believe that even considering that, particle deposition should lead to finer structures compared to evaporating droplets deposited by syringe. What could be the influence of a fast evaporating solvent the structure of the dried deposit? Did you do experiments with others solvents?

line 216 and Line 219: "We were largely able to reproduce our measurement results..." and "with practically no free parameters" , what do 'largely' and 'practically no' mean in this context, please, rephrase or specify.

line 326-329: This statement should be appear in the result section before.

Typos:

line 30: the these, cancel "the"

line 52: "stems", turn to plural

line 62: Bannan et al. 2019, add the brackest to year

line 70: "that atomizer method", add "the"

line 486(Figure 1): "for the further divergence", didn't you mean "larger divergence?

line S48: "d", should be "f"

---

## Referee Comment (RC3) · Anonymous Referee #3 · 3 Sep 2020

OVERALL

The authors have studied the how the means of administering calibrant compounds affects the calibration of the FIGAERO-CIMS desorption temperature vs the compound saturation pressure. The authors find that there is a possible source of error when administering the calibrate solution directly the FIGAERO filter via a syringe. They propose an alternative method based on aerosolizing the calibrant.

Because of the increasing use of the FIGAERO, the findings of the paper are important. The alternative methodology is useful and easy to apply for most of the users. The methodology appears solid and the overall clarity of the paper good, conveying the

message. I have some comments below, but overall, I recommend publication.

GENERAL

Modeling (Page 6, 3rd paragraph; page 7 last lines; page 8, 3rd and last paragraphs, page 11 1st paragraph): The model brings enough insight to justify itself for the paper, but I think it could be better tied to the results. As the model predicts the effect of particle diameter, maybe this aspect could be dealt with first. One could estimate the surface to volume ratio of the residue to that of the fitted 11 um diameter particles. It seems that an order of magnitude agreement would be achieved with a reasonable residue thickness estimate. Another thing: why is the model insensitive to particle diameter for PEG-8?

The authors might like to check for papers that have came out after preparing the manuscript.

SPECIFIC COMMENTS

Abstract:

There would be room for specific numbers on the effects of the methods on peak temperature, pressure and concentration.

Methods:

Page 4 and onwards: I understand the concentration probably affects the residual ring pattern, but it would be nice to also show the deposited solute mass.

Eq (3) and related texts: C*-space is referred to later in the text several times, as is customary to the field. On the other hand, this paper (or the supplement) does not provide the physical properties of the compounds needed to convert the saturation pressure values to concentrations. Including the equation is therefore of limited value. Maybe the properties could be listed in the supplement? Or then just point out here that this is a linear function. BTW: molecular weight/mass should be changed to molar

here, and later when appropriate.

Results and discussion

Figure 3 and related texts: Overall this is a nice set, conveying the message. I have some minor points: -Please harmonize the two panes for marker and text sizes -There would be more room to spell out the Saturation vapor pressure within the caption than on the axis. This is shortened to Saturation pressure elsewhere in the text. -I guess the lines are just to guide the eye. Why are they missing form the b) and why is the PEG atomizer data included? -I propose checking (dashed) horizontal lines to the Psat values -Please refer to supplement here for the Psat values -I did not find mention of the sources of the carboxylic acid Psat values. These should be added.

Figure 5: The scale bar text is too small. Although is implicitly clear, maybe the caption should spell out that this is after solvent evaporation. Maybe use the word residue?

Figure 6: I know there is little that can be done here, but it is practically impossible to tell the particles apart from the filter. The marked particles are approximately one micrometer in diameter, not 300 nm. Any explanation?

Page 8, last paragraph and figure 7: Explain what the difference in the inlet was, and why it affects the Tmax values.

Figure 8: Nice result, but as this is calibration, should there not be an error estimate?

Page 10, 2nd paragraph: "Note that the heating ramp rates in these calibrations were done with faster heating ramp rate. . ." This sentence can be shortened quite a bit. Apart from that, maybe the authors would like to discuss the effect of the ramp rate in the light of this paper. . .

---

## Author Comment (AC1) · 25 Oct 2020

Response to reviewer comments for manuscript: "On the calibration of FIGAERO-ToF-CIMS: importance and impact of calibrant delivery for the particle phase calibration"

Ylisirniö et al.,

We thank the reviewer for his/her constructive comments regarding our paper. Below we will address the specific issues point by point. The reviewer's comments are in black and our answers are in blue.
Changes to the Manuscript or Supplement Information are highlighted in red.
Line numbers before the red response text refer to line numbers in the modified manuscript.

**Additional changes by authors:**

We added additional discussion about potential effect of collected aerosol mass loading to Section 3.3.

Line 281:

"An additional aspect that has been reported to shift $T_{max}$ values is the amount of collected aerosol mass on the PTFE filter (Huang et al., 2018), becoming important when collected particulate mass is around several micrograms. We tested the mass loading effect by collecting different amounts of atomized PEG's up to 200 ng of mass and found no clear difference between measured $T_{max}$ values (data not shown). However, as collected aerosol mass on the FIGAERO filter can easily reach microgram amounts, especially when sampling in highly polluted environments and as we did not rigorously test how $T_{max}$ values behave above 200 ng, we suggest that this effect is investigated further in future publications."

**Reviewer 1:**

General comments:

1) Dependence on the solution concentration. The authors demonstrate that by adjusting the diameter in the evaporation model, they can reproduce the results for different solution concentrations. In order to have this in line with the subsequent reasoning about the SEM image, they should show different SEM images with different solution concentrations showing that indeed smaller structures are deposited on the filter in the case of lower solution concentrations. If the diameter really controls this behavior, also a calibration with the atomizer and particles around 1 µm could support this.

The reviewer makes a good point regarding the additional SEM figures with different solution concentrations. Below is a SEM figure showing deposition using a solution of 0.25 g L$^{-1}$ which is a much higher concentration than the 0.01 g L$^{-1}$ used in Fig. 5b) of the manuscript. Careful analysis

may reveal some differences between the "ring structures" in these two cases. But, in our opinion, it is almost impossible to do a quantitative analysis of the true size and mass of the evaporating unit from the filter based solely on this type of SEM pictures. While it is easy to measure the diameter and width of the "ring", we have no information about the thickness of the layer. Additionally, part of the deposited compound/mass may be hiding inside the filter and cannot be seen with this method.

Thus, it is our understanding that a more thorough analysis using a higher number of different solution concentrations would not lead to more relevant information. We will continue to just use the SEM pictures to qualitatively verify that the two deposition methods (syringe vs atomization) lead to very different structures of the deposited material on the filter which impact the evaporation behaviour. However, we will more strongly emphasise the qualitative nature of the SEM pictures in the modified manuscript.

Line 242:

"We want to emphasize that as SEM cannot distinguish deposited material situated inside the filter or measure the layer depth, the images shown here should be considered only as qualitative evidence."

Unfortunately, creating monodisperse particles in the range of 1 µm is not trivial with our equipment. Optimising the particle generation and size selection for that size range was outside of the scope of this study.

[Figure]

*Figure 1. Panel a) Shows Fig.5 b) from manuscript showing 3 µl of PEG-8 with concentration of 0.01 g $L^{-1}$. Panel b) shows SEM figure of 3 µl of PEG-8 with concentration of 0.25 g/l in ACN deposited on the filter.*

2) The authors base their reasoning mainly on the SEM images of the FIGAERO filter for the atomization method and the syringe method. However, they only show SEM pictures for one substance. While I don't doubt their conclusion, other SEM images should be added. The authors make the statement that the vacuum in the SEM could evaporate all the other substances than PEG-8, but how can they than conclude that it is not the SEM sample preparation (i.e. bringing the filter into a vacuum), which causes the structures observed on the filter?

We argue that the formation of the ring structure is caused by the evaporation of the initially deposited droplet, i.e., the evaporation of the solvent, ACN. Changing the rate of evaporation (i.e., by evaporating at ambient or SEM pressure) may change the exact size and shape of the "ring

structure", but the structure will still be formed and be very different from the same mass deposited as sub-micron particles.

The only difference between normal FIGAERO sample treatment and SEM sample treatment is that SEM is operated in vacuum while in the FIGAERO the samples are not exposed to lowered pressures. PEG-8 was selected for the screening due to its low vapour pressures, to minimise the likelihood of any evaporation of the example calibration compound in the vacuum of the SEM which might skew the results from the SEM. While it is possible to use some solid compound like citric acid, which would not easily evaporate from the filter even in vacuum, getting these SEM pictures would be a considerable effort, especially under the current circumstances with limited personnel available due to the pandemic situation.

3) Impact of using different calibration methods. When showing the different VBS systems, I would like to see also a comparison to a VBS derived using a group contribution method or a fit to it (as e.g. in Stolzenburg et al., 2018 or Mohr et al., 2019). This would indicate which calibration method is more in line with this widely used approach, which does not rely on a direct volatility measurement.

Determining the VBS distributions with group contribution methods and comparing them to VBS distributions determined using direct volatility measurements is indeed an interesting topic, but we decided to leave that analysis out from our manuscript as we think it to be a whole topic of its own. One important issue is that most of the parameterisations were developed for measurements of gaseous compounds. When a thermal desorption step is included, the extend of thermal decomposition must be considered, i.e. that low-volatile but thermally labile compounds are detected as small fragments which will lead to a significant overestimation of their $C*$ values (see e.g. Buchholz et al 2020, Lopez-Hilfiker et al. 2015, Schobesberger et al. 2018, Stark et al. 2017).

However, we are currently working on this topic and will discuss it in more detail in future publications.

**Minor:**

1) P.1, l.31-32: I am missing a short discussion on other volatility measurement techniques, e.g. VTDMA setups. Please add this here.

Added reference to VTDMA.

2) P.2, l.39: I am missing some laboratory studies from the CLOUD team published recently, e.g. Wang et al. (2020, Env. Sci. Techn. & Nature), Stolzenburg et al. (2018, PNAS). Also missing is Mohr et al. (2019, Nat. Commun.). In all these studies the FIGAERO-CIMS was deployed quite successfully and they could be mentioned here for completeness.

Added Stolzenburg et al. 2018 and Mohr et al. 2019 to references listed at this line and additionally also added Wang et al. 2020 to the reference list of published calibration lines.

3) P.2, l.49: Also Wang et al. (2020, Env. Sci. Techn.).

Added Wang et al. 2020 to reference list, and updated Figure 1 and Figure S2 with their calibration line.

4) P.5, l.144: Did you constrain the width of the lognorm fit for the desorption? This could be necessary especially for unknown compounds, which might have isomers or fragments on the same mass yielding a bimodal structure.

The reviewer is right that often the fit needs to be constrained. As the calibration compound thermograms are "ideal", lognormal fit can be applied to the whole thermogram, but in "real" data the fit usually needs to be constrained around the peak of the thermogram, especially if the thermogram is bimodal or broadened by the presence of isomers and/or fragments from thermal decomposition. Note that in the case of the presence of multiple compounds of different volatility, the $T_{max}$ value may represent only the volatility of the dominant compound and ignores the contribution of the minor compounds with that sum formula. But it is also possible that the $T_{max}$ value represents an "average" over multiple compounds, especially if the thermogram peaks of the isomers/fragments are too close to be distinguished. For such cases, the $T_{max}$ method cannot capture all of the volatility information and a more sophisticated method is needed to separate the compounds (e.g. Positive Matrix factorisation, Buchholz et al. 2020)

5) P.6, l.185-187: If the inlet is initially at a different temperature, the supply of a constant heat rate will yield a different thermogram, as it takes longer to achieve the corresponding temperatures allowing more time for evaporation. Is this considered in then model? And how can we use calibrations performed at one temperature in comparison to measurements at different temperatures? Could the model resolve this?

The FIGAERO uses filters made of only PTFE. The filter holder and the moving tray are also machined from PTFE. One reason for that choice is the material's chemical inertness. But the main reason for choosing PTFE is its low thermal conductivity. Consequently, the temperature of the deposit is more directly controlled by the heat of the N2 flowing through the filter, which is measured immediately upstream and hence well understood. The model therefore does not explicitly consider the heating of the filter material. But it does allow for "non-ideal" heating of certain parts of the deposit. The description of that non-ideality is fairly crude (details in Schobesberger et al., ACP, 2018). It has not been modelled as a function of heating rate, as that has not appeared necessary. The main "job" of the non-ideal heating in the model is to produce thermogram tails; it hardly affects $T_{max}$.

If the reviewer's latter two questions refer to the *ambient* temperature of the calibrations, the model is not currently set up to resolve resulting issues.

If the reviewer refers to extending the experimentally obtained "calibration curve" to $T_{max}$ values beyond those provided by the observed $\underline{T_{max}}$ (illustrated e.g. in Fig. S2, and subject of Fig. S3), the model could in principle do that, but it would require asserting a relationship between saturation vapor pressures and vaporization enthalpies (e.g., see Fig. 7 in Schobesberger et al., 2018).

6) P.7, l.202: Repeat the atomizer solution concentration to put it into the context with the syringe concentrations. Also mention here the mode diameter of the particles used for calibration or even calculate the deposited mass for this type of calibration compared to the syringe method. This would put the two methods into comparison here.

Added more information to the section.

Line 208 onwards:

"For comparison, the starting concentration of the atomizer solution was 0.5 g L-1 for each compound. The solution concentration gradually increased as the solvent evaporated from the solution. This led to a polydisperse, log-normal-shaped aerosol population with a mode diameter of 50 nm. From this distribution, particles equivalent to ~200 ng of aerosol mass were sampled onto the FIGAERO filter before desorption."

7) P.8, l.231: Instead of mentioning the different scale, I would like to see a fourth panel in Fig 4 showing the filter in the same scale as in Fig. 5c! This would help to directly compare the different structures deposed on the filter.

Added fourth panel to Figure 5 (panel c) with 10 μm scale, taken as same sample as Fig. 5a). Original Fig. 5c) is now Fig 5d).

8) P.8, l.240: Also the larger diameters needed to explain the syringe calibration with model point into that direction. This is an important supportive argument and should be mentioned here.

We added the following to the revised manuscript:

Line 254 onwards:

Indeed, it was by building on these assumptions that the evaporation model succeeded in reproducing the observations in Fig. 4. With the much smaller surface area of the syringe deposited material, it requires more time to evaporate all the PEG-8 than from the equivalent amount of deposited aerosol particles. This time delay directly translates to a shift to higher observed $T_{max}$ values. The desorption model mimics this change in surface-to-volume ratio by increasing the initial size of the modelled evaporating particle to 1.3 μm and 11 μm. But note that there are no individual spherical particles of that size on the filter.

9) P.8, l.251: Any hints why the different inlet behaves that way?

The positioning of the filter thermocouple affects the measured temperature of the desorption flow. It also affects the exact offset between the measured temperature and the temperature at the filter surface. To position the thermocouples in exactly the same position in two different inlets is quite difficult, so it is relatively hard to achieve exactly comparable measurements using separate inlets.

This is one of the reasons why a temperature calibration is necessary for each FIGAERO inlet and also every time the inlet is disassembled as disassembling the inlet may affect the position of the thermocouple.

10) P.8, l.254: Why does it fail for PEG-8? Please elaborate on that.

In the model simulation, molecules that have desorbed from deposited particles are subsequently interacting with instrument surfaces, which are experiencing the same temperature ramp as the deposit, before being measured. That interaction is simulated by 100% initial absorption, and desorption as a function of $C*(T)$ and an optional instrument constant. (The latter is indeed the main tuning parameter for the model to reproduce observed $T_{max}$ values). These interactions cause a delay. The delay translates to higher $T_{max}$ and increases with decreasing $C*$ as well as decreasing particle size. Consequently, differences in $T_{max}$ due to particle size disappear when $C*$ and particle size are sufficiently small, as $T_{max}$ becomes controlled by those vapor-surface interactions. This is what we observe in the model outputs for PEG-8 (and somewhat also for PEG-7).

We believe that that is a shortcoming of the model, rather than of the experiments, because: (a) the thermograms were experimentally very well reproducible, and (b) the thermograms (in particular for PEG-8) were narrower for 80-nm particles than for 300-nm particles. From the model's point of view, observation (b) would suggest more ideal heating in the 80-nm case than in the 300-nm case, while the lower $T_{max}$ would suggest less vapor-surface interactions. To affect that, 80-nm particles would need to be deposited somehow substantially differently on the filter than 300-nm particles. More likely instead, the model's current treatment of both non-ideality of heating and vapor-surface interactions (Schobesberger et al., 2018) are insufficiently close to reality in this case.

11) P.10, l.295: Move "A more detailed description of the SOA production is shown in Ylisirniö et al., 2020." in front of the preceding sentence.

Sentence moved.

12) P.10, l.299: 200 g mol-1 seems quite low for alpha-pinene HOMs, e.g. Tröstl et al. used 300 amu as mean mass.

The reviewer is correct that 200 g mol$^{-1}$ is small compared to the mean molecular mass of alpha-pinene HOMs. However, when $C*$ is plotted in logarithmic space as in Fig.9, the change from 200 g mol$^{-1}$ to 300 g mol$^{-1}$ becomes negligible, as can be seen in the figure below. We therefore think that 200 g mol$^{-1}$ is adequate for our purposes.

[Figure]

*Figure 2. Figure 9 panel d) syringe deposition method theoretical line calculated with 200 g mol⁻¹ and 300 g mol⁻¹.*

13) P.11, l.355: Seems logical, but extremely difficult to realize in the lab. What would be the best alternative?

We are not completely sure what the reviewer means with this question as the sentence in the line 355 reads: "We note that these $P_{sat}$ values have not been verified by other studies and are subject to corrections, but want to point out that harmonizing further FIGAERO calibrations by using PEGs would make future FIGAERO measurements more comparable to each other."

We want to note that by word "harmonizing" we don't mean a rigorous ISO-standard style calibration procedure, but simply that each FIGAERO would be calibrated with same compounds and using same $P_{sat}$ values.

---

## Author Comment (AC2) · 25 Oct 2020

Response to reviewer comments for manuscript: "On the calibration of FIGAERO-ToF-CIMS: importance and impact of calibrant delivery for the particle phase calibration"

Ylisirniö et al.,

We thank the reviewer for his/her constructive comments regarding our paper. Below we will address the specific issues point by point. The reviewer's comments are in black and our answers are in blue. Changes to the Manuscript or Supplement Information are highlighted in red.
Line numbers before the red response text refer to line numbers in the modified manuscript.

**Additional changes by authors:**

We added additional discussion about potential effect of collected aerosol mass loading to Section 3.3.

Line 281:

"An additional aspect that has been reported to shift $T_{max}$ values is the amount of collected aerosol mass on the PTFE filter (Huang et al., 2018), becoming important when collected particulate mass is around several micrograms. We tested the mass loading effect by collecting different amounts of atomized PEG's up to 200 ng of mass and found no clear difference between measured $T_{max}$ values (data not shown). However, as collected aerosol mass on the FIGAERO filter can easily reach microgram amounts, especially when sampling in highly polluted environments and as we did not rigorously test how $T_{max}$ values behave above 200 ng, we suggest that this effect is investigated further in future publications."

**Reviewer 2**

**Major comment:**

The manuscript deals with calibration issues, which means it deals with quantitative issues, and it compares to results from other references. Although the authors made aware of suited fitting procedures for data with errors in y and x (line 168f) there is no detailed error analysis nor are error bars shown in the Figures 1, 4, 7, 8, S2, S3. The only errors given are the statistical errors from averaging single measurements. I find that strange for a paper that deals with quantitative analysis and urgently suggest to add more detailed error analysis' and -discussion.

The reviewer is correct that more detailed description about the errors is needed. The main results of the paper are the $T_{max}$ value measurements, i.e. the actual temperature measurements and its associated errors. The FIGAERO inlet uses two k-type thermocouples, which have typical measurement error of +/- 2.2 K. Additional ~1K of error is also introduced by the electronics of the

measurement system. However, this total uncertainty of ~3 K can be thought of as a systematic error, rather than a random error. Therefore, the calibration procedure itself captures this uncertainty. Any random component of the temperature measurement error is then captured with repetitions and shown in standard deviations. An additional component to the error is introduced in the fitting of the asymmetric lognormal function over the measured thermograms. This error can be estimated by using the fitting routine's goodness-of-the-fit parameters, such as R-square value.

In Figures 1, and S2 the error bars are omitted for sake of clarity, but this is now pointed out in the captions and errors are discussed in the sect. 2.5.

Error bars and their description are now added to Figure 4 and explained in the caption.

In Figure 7, the error bars are omitted from the figures measured values as they are not distinguishable in the x-axis direction and do not add value to the plot. A range of the models results are however now shown in panel b) when using the uncertainties for evaporation enthalpy ΔH shown in Krieger et al (2018). The y-axis error values are the same as in Figure 4. The range of the errors is now given in the figure caption:

"Error bars are omitted from the measured values in the figure for sake of clarity. In the panel b) whiskers show the range of model results when using uncertainties of evaporation enthalpy shown in (Krieger et al., (2018). The standard deviations of the for all measured points is panel a) is between 0.2-0.5 ˚C. In panel b) the standard deviations for measured points are between 0.2 – 1.3 ˚C."

For Figure 8 error bars are now shown in the plot and the error analysis is explained in the supplement information S4. Briefly, the x-axis errors are calculated with propagation of error for both syringe injections and aerosol collections. The Y-axis errors are estimated by assuming Poisson-type counting statistics.

In Figure S3 the error bars are not shown as the idea of the figure is to be mainly speculative on how the $P_{sat}$ values of higher order PEG-compounds could possibly be estimated.

**Minor comments:**

line 44: References should be given already here, in addition to the link to section 2.5.

Moved citations.

line 82: Are these heating rates really so accurate (2 decimal digits)? The experiments do not lead to the same maximum temperature, does that effect the integrals under the thermograms?

We decreased the accuracy of the ramp rate to one digit. For the integration, the thermogram is integrated against time and not temperature, as data is measured in counts per second. Therefore, maximum attained temperature does not affect the integral value as long as it was high enough for all material to desorb from the filter.

line 97: From PEG-4 to PEG-8. In integer steps? Please, specify precisely which PEG's you used.

Text has been modified to mention all used PEG-compounds.

Line 99:

The used PEG standards were PEG-4, PEG-5, PEG-6, PEG-7 and to PEG-8.

line 115-120: Did you use a neutralizer ? If yes, which? How did you handle multicharged particles in the selection by DMA. This needs to be explained, and potential errors need to be estimated and discussed.

The TSI model 3082 SMPS-platform contains a Kr-85 radioactive neutralizer and the measurement software automatically corrects for multicharged particles. When used in DMA mode for monodisperse aerosols, the particle size was selected from the falling edge of the polydisperse particle distribution, keeping the amount of multicharged particles at a minimum. For example, the measured mode diameter of the polydisperse aerosol distribution was ~50nm and we selected 100 nm monodisperse particles for the sensitivity measurements.

Figure 6: I don't see a real difference between with and w/o particles. What made you think that the marked white blobs are particles? The marking with the red circles is too suggestive. I propose to leave them out. In this context: In the ambient with aqueous particles and RH, wouldn't the particles flow together and merge anyhow? I would still believe that even considering that, particle deposition should lead to finer structures compared to evaporating droplets deposited by syringe. What could be the influence of a fast evaporating solvent the structure of the dried deposit? Did you do experiments with others solvents?

The reviewer is right that it is indeed difficult to distinguish the PEG-8 particles from the filter material. We base our estimation on the quantity of the white "blobs" seen in the picture with deposited particles compared to the clean filter.

When considering aqueous particles, the distance between the collecting filter fibres compared to the size of the collected aerosol particles is large enough that the particles should not meet each other unless using very high mass loading when collecting the sample. The aerosol particles are also so small that intermolecular forces between the particles and filter fibres should be strong enough to prevent them from moving around in the filter by the collection flow drag. Note that for high sample mass on filters, sample mass dependent shifts of $T_{max}$ have been observed (Huang et al. 2018, Atmos. Chem. Physics) which may be connected to particles interacting on the filter.

As the PEG-8 molecules are floating freely in the acetonitrile (ACN) solution without diffusion limitations, the evaporation rate of the solvent should not affect how the PEG-8 molecules deposit onto the filter.

As for other solvents, to dissolve PEGs a polar solvent is needed. However, we observed oligomerisation reaction for PEG4 when using alcoholic solvent (methanol). In addition to ACN we did test ethyl acetate and deionized water as solvents for both the syringe and the atomizer method but did not take SEM pictures with these solutions. We decided to use ACN instead of Ethyl Acetate as its less volatile. The higher volatility of ethyl acetate could have caused problems in

atomization as the atomizer solution concentration would have changed too rapidly as the solvent evaporates. Additionally, the PTFE-filter is less phobic towards ACN than it is towards water, which eases the syringe deposition experiments. For these reasons, we recommend using ACN as solvent for PEGs. However, if ACN is not available, it is possible to replace ACN with another aprotic/non-alcoholic, somewhat polar solvent at least in the atomizer method.

line 216 and Line 219: "We were largely able to reproduce our measurement results: : :" and "with practically no free parameters" , what do 'largely' and 'practically no' mean in this context, please, rephrase or specify.

We specified and tried to clarify those two sentences, now reading as follows:

Line 228 onwards:

"We were able to reproduce our measured $T_{max}$ values within 10 °C using the evaporation model to simulate the evaporation of mixed PEG 4-8 particles (for simplicity assuming equal mole fractions for all PEG). For PEG-5 and -6, Figure 4 shows excellent agreement between measured and modelled $T_{max}$ values for the atomizer method (within a couple of °C), deteriorating to a difference of about 10 °C for PEG-8. This broad agreement here is remarkable in so far, as in this case the model was run with no vapor-surface interactions, i.e. no tuning in regards to resulting $T_{max}$, which are therefore a direct result of the input values for C* and ΔH."

line 326-329: This statement should be appear in the result section before.

Moved the paragraph to Section 3.1.

**Typos:**

line 30: the these, cancel "the"

Done.

line 52: "stems", turn to plural

Done.

line 62: Bannan et al. 2019, add the brackest to year

Done.

line 70: "that atomizer method", add "the"

Done.

line 486(Figure 1): "for the further divergence", didn't you mean "larger divergence?

Replaced with "large":

"…for the large divergence…"

line S48: "d", should be "f"

Done.

---

## Author Comment (AC3) · 25 Oct 2020

**Response to reviewer comments for manuscript: "On the calibration of FIGAERO-ToF-CIMS: importance and impact of calibrant delivery for the particle phase calibration"**

**Ylisirniö et al.,**

We thank the reviewer for his/her constructive comments regarding our paper. Below we will address the specific issues point by point. The reviewer's comments are in black and our answers are in blue. Changes to the Manuscript or Supplement Information are highlighted in red.
Line numbers before the red response text refer to line numbers in the modified manuscript.

**Additional changes by authors:**

We added additional discussion about potential effect of collected aerosol mass loading to Section 3.3.

Line 281:

"An additional aspect that has been reported to shift $T_{max}$ values is the amount of collected aerosol mass on the PTFE filter (Huang et al., 2018), becoming important when collected particulate mass is around several micrograms. We tested the mass loading effect by collecting different amounts of atomized PEG's up to 200 ng of mass and found no clear difference between measured $T_{max}$ values (data not shown). However, as collected aerosol mass on the FIGAERO filter can easily reach microgram amounts, especially when sampling in highly polluted environments and as we did not rigorously test how $T_{max}$ values behave above 200 ng, we suggest that this effect is investigated further in future publications."

**Reviewer 3:**

**GENERAL**

Modeling (Page 6, 3rd paragraph; page 7 last lines; page 8, 3rd and last paragraphs, page 11 1st paragraph): The model brings enough insight to justify itself for the paper, but I think it could be better tied to the results. As the model predicts the effect of particle diameter, maybe this aspect could be dealt with first. One could estimate the surface to volume ratio of the residue to that of the fitted 11 μm diameter particles. It seems that an order of magnitude agreement would be achieved with a reasonable residue thickness estimate. Another thing: why is the model insensitive to particle diameter for PEG-8? The authors might like to check for papers that have came out after preparing the manuscript.

We believe that the storyline works well enough as it is, and it is more in line with how our understanding progressed during this study. We first had the observation of increased desorption

temperatures for syringe methods, and blaming that on reduced S:V was one of several initial hypotheses, but the one that crystallized as the most plausible explanation. Our evaporation model is able to simulate that effect, in its simplifying approach, but nonetheless able to roughly quantify it, so we agree that the obtained numbers are providing useful "ballparks". (The SEM pictures were then taken last.)

Estimating the S:V of the residue, based on the SEM pictures, is an interesting approach – if very qualitative only, as the "reasonable residue thickness estimate" is a major uncertainty. But if we anyway, for example, looking at Fig. 5, maybe a good guess would be a residue consisting of 20x20 $\mu m^2$ blocks with a depth of 1 $\mu m$? The corresponding S:V is 2.2 $\mu m^{-1}$, equivalent to the S:V of a sphere (particle) of diameter 2.7 $\mu m$. Indeed, a reasonable order of magnitude agreement is achieved (the model using 1.3 $\mu m$; Fig. 4). But a major problem remains with the lack of depth information obtainable from the SEM images, as up to the filter thickness (~150 $\mu m$), a suitable residue thickness can mostly be found ("estimated") to come up with the particle size value desired. Indeed, one could argue that, vice versa, SEM images plus modelled particle size could be used to estimate residue thickness. Adding to that argument are the similarities regarding apparent residues between Fig. 1 panels a) and b) in this document above, despite different solution concentrations used (0.01 and 0.25 g/L, respectively), whereas the model would use particle sizes of 1.3 and >11 $\mu m$, respectively (Fig. 4). But if we keep with 20x20 $\mu m^2$ patches, we only need to increase their depth, for example to 5 $\mu m$, and the corresponding particle size would increase to 10 $\mu m$. Voilà. So those calculations are able to provide reasonable numbers. That is encouraging, but we feel that they remain poorly constrained by our observations (plus model simulations), and we do not learn much more, and only with high uncertainties.

Regarding PEG-8/Fig. 7, please see response to Reviewer 1, comment 10.

Regarding the new papers that have been published during the review progress, we have added Wang et. al., (2020) calibration line to Figure 1 and Figure S2.

**SPECIFIC COMMENTS**

Abstract:

There would be room for specific numbers on the effects of the methods on peak temperature, pressure and concentration.

Added an example about the difference in $T_{max}$ between atomizer method and syringe method and how big effect this would cause in C*space.

Line 22:

"For example, we found a difference of ~15 ˚C in observed $T_{max}$ values between the atomizer method and the syringe method when using the lowest solution concentration (0.003 g L-1). This difference translates to up to 3 orders of magnitude difference in saturation concentration $C*$ space."

Methods:

Page 4 and onwards: I understand the concentration probably affects the residual ring pattern, but it would be nice to also show the deposited solute mass.

Deposited solute masses are already being reported in the end of the Sect. 2.3. "The mass deposited varied between 9 ng and 500 ng, depending on the used concentration." In the SEM pictures amount of deposited mass was 30ng. This information is now added to the figure caption.

Eq (3) and related texts: C*-space is referred to later in the text several times, as is customary to the field. On the other hand, this paper (or the supplement) does not provide the physical properties of the compounds needed to convert the saturation pressure values to concentrations. Including the equation is therefore of limited value. Maybe the properties could be listed in the supplement? Or then just point out here that this is a linear function. BTW: molecular weight/mass should be changed to molar here, and later when appropriate.

If the reviewer refers to $P_{sat,meas}$ or $M_w$ as physical properties of the compounds, the $P_{sat,meas}$ value is determined from measured $T_{max}$ value using eq(2) and $M_w$ is determined directly from the CIMS data. This is now clarified in the text and molecular weight is changed to molar mass.

Line 164:

"where $M_w$ is the molar mass of the compound (in units of g mol$^{-1}$) determined with the CIMS,"

**Results and discussion**

Figure 3 and related texts: Overall this is a nice set, conveying the message. I have some minor points: -Please harmonize the two panes for marker and text sizes -There would be more room to spell out the Saturation vapor pressure within the caption than on the axis. This is shortened to Saturation pressure elsewhere in the text. -I guess the lines are just to guide the eye. Why are they missing form the b) and why is the PEG atomizer data included? -I propose checking (dashed) horizontal lines to the Psat values -Please refer to supplement here for the Psat values -I did not find mention of the sources of the carboxylic acid Psat values. These should be added.

Figure 3 is now modified as requested. Used $P_{sat}$ values for carboxylic acids and their reference are listed in Table S1 and this is mentioned in the text.

The lines connecting the dots in Fig. 3a) are indeed to guide the eye. They are however missing from Fig 3b) from carboxylic acids as the data is more compacted and thus would make the figure harder to read. Also, the PEGs are a homologous series, whereas the acids are not. The PEG atomizer data is included for easier comparison between the two panels as the x-axis is different in each panel.

Figure 5: The scale bar text is too small. Although is implicitly clear, maybe the caption should spell out that this is after solvent evaporation. Maybe use the word residue?

Increased the scale bar text and added fourth panel as suggested by Reviewer 1. New panel shows zoomed in picture of panel a). Changed the word 'PEG-8 "ring"' to 'PEG-8 residue'.

Figure 6: I know there is little that can be done here, but it is practically impossible to tell the particles apart from the filter. The marked particles are approximately one micrometer in diameter, not 300 nm. Any explanation?

It is indeed hard to distinguish the collected particles from the filter material. We base our estimation on the quantity of the white "blobs" seen in the picture with deposited particles compared to the clean filter. The amount of material collected onto the filter was substantially higher than what would be normally collected during experiments, to ensure the identification of the particles from the filter material. It is possible that some of particles have coagulated in the filter and formed bigger particles.

Page 8, last paragraph and figure 7: Explain what the difference in the inlet was, and why it affects the Tmax values.

The other inlet used in this paper was a slightly modified version from the general Aerodyne inlet. The two inlets differ geometrically slightly from each other in terms of pin hole and thermocouple positioning. However, as the point of the calibration is to ensure that different FIGAERO systems are comparable to each other, slight changes in the inlet geometry are not crucial. The largest source of error regarding $T_{max}$ values comes from the exact positioning of the filter thermocouple, which is individual to each FIGAERO inlet, and is an important reason to use harmonised calibration method between different FIGAERO instruments.

Figure 8: Nice result, but as this is calibration, should there not be an error estimate?

Error bars are now added to the figure and the error analysis is explained in the supplement information.

Page 10, 2nd paragraph: "Note that the heating ramp rates in these calibrations were done with faster heating ramp rate…" This sentence can be shortened quite a bit. Apart from that, maybe the authors would like to discuss the effect of the ramp rate in the light of this paper…

We modified the sentence:

Line 317:

"Note that the calibrations shown in this paper used faster heating ramp rates than what was used in the SOA measurements, introducing an overall small systematic error (<1 order of magnitude in $C^*$ space) towards higher saturation concentrations."

The effect of the used ramp rate on the observed $T_{max}$ values is discussed in Sect. 3.3, last paragraph, and the effect is shown in Figure 7.